# A Query Expansion Benchmark on Social Media Information Retrieval: Which Methodology Performs Best and Aligns with Semantics?

Evangelos A. Stathopoulos *[ID], Anastasios I. Karageorgiadis, Alexandros Kokkalas, Sotiris Diplaris [ID], Stefanos Vrochidis [ID] and Ioannis Kompatsiaris [ID]

Information Technologies Institute, CERTH, Thermi, GR 57001 Thessaloniki, Greece; tassoskarag@iti.gr (A.I.K.); akokkalas@iti.gr (A.K.); diplaris@iti.gr (S.D.); stefanos@iti.gr (S.V.); ikom@iti.gr (I.K.)
* Correspondence: estathop@iti.gr

**Abstract:** This paper presents a benchmarking survey on query expansion techniques for social media information retrieval, with a focus on comparing the performance of methods using semantic web technologies. The study evaluated query expansion techniques such as generative AI models and semantic matching algorithms and how they are integrated in a semantic framework. The evaluation was based on cosine similarity metrics, including the Discounted Cumulative Gain (DCG), Ideal Discounted Cumulative Gain (IDCG), and normalized Discounted Cumulative Gain (nDCG), as well as the Mean Average Precision (MAP). Additionally, the paper discusses the use of semantic web technologies as a component in a pipeline for building thematic knowledge graphs from retrieved social media data with extended ontologies integrated for the refugee crisis. The paper begins by introducing the importance of query expansion in information retrieval and the potential benefits of incorporating semantic web technologies. The study then presents the methodologies and outlines the specific procedures for each query expansion technique. The results of the evaluation are presented, as well as the rest semantic framework, and the best-performing technique was identified, which was the curie-001 generative AI model. Finally, the paper summarizes the main findings and suggests future research directions.

**Keywords:** query expansion; social media information retrieval; generative AI models; semantic matching algorithms; semantic web; ontology; knowledge graph

## 1. Introduction

Social media platforms have become a vital source of information in today's digital age. They cater to billions of users and generate enormous amounts of data daily. Consequently, effectively retrieving relevant information from these platforms is no longer a trivial task, but a significant challenge, underscoring the importance of effective Information Retrieval (IR) techniques.

The global crisis of forced migration and the plight of refugees is a critical issue that affects millions of people worldwide. Cultural institutions, including museums, galleries, and archives, have a significant role to play in raising awareness about this issue, promoting empathy and understanding among the public [1,2]. However, creating immersive digital experiences about forced migration requires a comprehensive understanding of the complex social, political, and economic factors that contribute to forced displacement and the challenges that refugees face in their daily lives [3].

To address this challenge, there is a need for high-quality information and data about forced migration and refugees. This information can help cultural institutions develop immersive digital experiences that accurately represent the experiences of refugees and create meaningful connections with audiences. Gathering this information is a complex

and multifaceted task that requires a range of different research methods and data sources, including academic research, fieldwork, interviews, and social media analysis [4,5].

By combining these different approaches, cultural institutions can create a comprehensive understanding of forced migration and refugees and develop immersive digital experiences that promote empathy, understanding, and social change. This paper aimed to contribute to this goal by presenting a survey of different techniques and methodologies in order to evaluate the best algorithms and tools about expanding queries in social media information retrieval to fuel a semantic web framework. The main goal was to detect the best combination of components that will result in the most-relevant multimodal data from Twitter and YouTube regarding the aforementioned topics. The derived social data must be semantically annotated in the most-efficient manner so as to be populated inside a knowledge graph [6]. Several information retrieval mechanisms, residing on top of the knowledge graph, are in charge of offering knowledge to the cultural institutions. The ultimate goal was to offer raw material, original user-generated content, and knowledge. This content was utilized to create immersive experiences about refugee and migrant stories, to try to connect the past with the present and bring people from so far so close. By raising awareness of the difficulties forced migrations include, local communities embark to become more inclusive as host societies to people deriving from different backgrounds, battling in that way racism, discrimination, marginalization, and ghetto formation.

Query expansion, a prevalent IR technique, enhances the efficacy of search results by appending relevant terms to the user's query, aiming to bridge the gap between user intention and query expression. With the advent of semantic web technologies, combining query expansion technique with semantic frameworks has gained considerable attention, promising improved search result relevance and user satisfaction.

However, this interplay of query expansion and semantic web technologies in the context of social media information retrieval is a comparatively uncharted territory. Existing research has either focused on traditional query expansion methods or the application of semantic technologies, with a lack of emphasis on their combined effect, specifically in the realm of social media.

In response to this research gap, this paper presents a comprehensive survey of query expansion methods used in social media information retrieval, focusing on methodologies synergizing with semantic web technologies as a sequential component in a pipeline populating a queryable knowledge graph. We rigorously evaluated the performance of various query expansion techniques, including emerging generative AI models such as GPT v3.0 and semantic matching algorithms retrieving synonyms and similar phrases from well-established public knowledge graphs such as DBpedia [7] and WordNet [8]. The techniques were evaluated by using cosine similarity measures to calculate the DCG value and IDCG to calculate the nDCG [9]. Additionally, the MAP was considered in order to provide a comprehensive evaluation of the query expansion techniques.

The objective of this survey was to identify an optimal query expansion methodology for social media information retrieval applied sequentially with semantic web technologies. We compared the results from different techniques, scrutinized their impact on various parameters, including the custom ontology used, the relevant content of the knowledge graph, and the number of expansion terms, and investigated the effectiveness of generative AI models versus semantic matching algorithms in query expansion.

The remainder of this paper is structured as follows. Section 2 provides an overview of related work in the field of query expansion in social media information retrieval and semantic web technologies. Section 3 unveils the frameworks developed during this study, while Section 4 explains the experimental setup and methodology employed for evaluating the performance of different techniques. Section 5 discloses the results of our experiments and analyzes the impact on the semantic framework. Finally, Section 6 elaborates a discussion on the methodology proposed and the findings of this paper, while Section 7 concludes the paper, providing limitations faced and directions for future work.

## 2. Related Work

Query expansion in social media information retrieval and semantic web technologies are two important areas in the field of the Worldwide Web. In this section, a provision of an overview of related work in these areas can be found that categorizes the literature into two broad areas.

Information retrieval systems facilitate the sourcing of applicable documents, such as web pages, to correspond with the demands of the end-users [10]. The classic methods used by these systems categorize documents by matching them to user inquiries, subsequently presenting the related documents to the users [11]. However, the indexing process is sophisticated and time-consuming, attributed to the large and varied nature of documents. Hence, standard information retrieval often struggles with imprecise search results and possibly diminished productivity. In addition, due to the textual interpretation limitations of search engines, keyword-based searches typically yield restricted outcomes. To mitigate these issues, contemporary search engines are increasingly employing knowledge graphs [12,13]. The use of knowledge graphs in information retrieval signifies a pioneering research trajectory aimed at enhancing search engine performance and the comprehensibility of their results.

The prevalent practice among these systems is to leverage a sophisticated document representation based on entities and their interrelationships derived from knowledge graphs. These structured, machine-understandable representations are then juxtaposed with user inquiries to source more relevant documents. For instance, Reference [14] introduced a COVID-19 Knowledge Graph (CKG) to discern the relationships among scientific articles on COVID-19. Specifically, they amalgamated the topological information of documents with their semantic connotations to devise document knowledge graphs. Another example is Wang et al. [15], who designed a knowledge-graph-centered information retrieval methodology that extricates entities through the exploitation of entity data on web pages, deploying an open-source relationship extraction method. Subsequently, the entities with relationships are interconnected to create a knowledge graph.

Knowledge graphs also prove beneficial for query expansion techniques, which serve to enhance user inquiries by incorporating pertinent concepts (such as synonyms) [16]. Query expansion is a technique used in IR to improve the relevance of search results by adding additional terms or phrases to the original query. The goal is to broaden the scope of the query and to capture additional documents that might be relevant to the user's information needs. For instance, Reference [17] showcased an Entity Query Feature Expansion (EQFE), which augments queries via the query knowledge graph, including structured attributes and text. Reference [18] put forward the Entity-Duet Neural Ranking Model (EDRM). EDRM combines the semantics derived from knowledge graphs with the distributed representations of entities in inquiries and documents. Following this, it sequences the search outcomes utilizing interaction-based neural ranking networks.

One other approach to query expansion is to use generative AI models [19], such as GPT-3, to provide synonyms of phrases and keywords in the original query. GPT-3 is a powerful language model that has been trained on a large corpus of text and can generate human-like text. Several studies have explored the use of GPT-3 for query expansion and have reported promising results. For example, Reference [20] proposed a query expansion method based on GPT-3 for microblog retrieval and showed that it outperformed several baseline methods. Similarly, a recent study by [21] used GPT-3 for query expansion in the context of biomedical literature retrieval and demonstrated its effectiveness.

Another approach to query expansion is to use semantic matching algorithms [22] to retrieve synonyms and similar phrases and keywords from well-established public knowledge graphs such as DBpedia [7] and WordNet [8]. Semantic matching algorithms can identify relevant concepts and entities that are related to the original query and expand the query accordingly. Several studies have explored the use of semantic matching algorithms for query expansion and have shown significant improvements in retrieval performance. For example, Reference [23] proposed an improved semantic expansion

method for short text retrieval based on ConceptNet, which outperformed several baseline methods. Similarly, Reference [24] proposed a query expansion method based on a hybrid model combining semantic similarity and text clustering techniques, which was shown to be effective in the context of product search.

IR does not guarantee homogenization; thus, semantic web technologies solve this problem by providing a framework for representing, sharing, and connecting data on the web, enabling the creation of a web of linked data where information is machine-understandable. The Resource Description Framework (RDF) [25] and Web Ontology Language (OWL) [26] are two of the key technologies used in the semantic web.

RDF enables the creation of rich and interconnected data graphs, while OWL provides a powerful ontology language for creating and sharing knowledge representations that define concepts and relationships between them. In the context of social media IR, semantic web technologies can be used to build thematic knowledge graphs that unify and organize the retrieved data from social media. The aforementioned knowledge graphs can be used to support various IR tasks, such as entity search and event detection. To query and analyze semantic data, standardized query languages such as the SPARQL Protocol and RDF Query Language (SPARQL) [27] were developed.

Recent studies have explored the use of semantic web technologies in various IR tasks, including social media IR. For instance, ontologies have been used to model event-related entities and relationships [28], while OWL ontologies were used to represent and organize entity-related information [29]. Semantic web technologies have also been used to support context-aware IR, personalized search, and recommendation systems [30]. In [31], unique query expansion techniques involve deriving a query's geographical footprint, accounting for spatial and non-spatial terms, the semantics of spatial relationships, and the context of use.

Furthermore, recent advances in Natural Language Processing (NLP) and machine learning have led to the development of novel approaches that leverage semantic web technologies to improve IR performance. For example, graph-based ranking algorithms that use semantic similarity measures are shown to improve retrieval performance in various domains, including social media [32]. Generative language models, such as GPT-3, were also explored for query expansion in the context of semantic-web-based IR [33].

## 3. Frameworks

During this study, two distinct frameworks were developed, which were connected sequentially to achieve targets, On the one hand, there is the information retrieval framework consisting of multiple tools, whose purpose is to gather multimodal multimedia from online sources. On the other hand, the semantic framework is in charge of homogenizing and semantically unifying the data from online sources along with other data (atomic and complex content items, which will be analyzed below) flowing inside the system to form the final knowledge graph. The latest framework also includes an additional component, the semantic search tool, to navigate and retrieve knowledge from the graph.

### 3.1. Information Retrieval Framework

The framework presented in this study is comprised of three sub-components: the Twitter crawler, the YouTube crawler, and the website-focused crawler. Each sub-component is designed to retrieve key data with common semantic properties, as well as unique information per source, in order to enhance the ontological integration of the data. The utilization of social media platforms to gather data on forced migrations and refugees is a useful approach, as these platforms generate continuous big data flows from users. The thematic content on these platforms can vary in modality, context, and metadata, making it necessary to monitor these flows in order to aggregate datasets and gain insights into various aspects of forced migration.

In this study, Twitter and YouTube were chosen as the best options for data collection, as they provide textual and audiovisual content that can greatly contribute to the production

of immersive digital experiences. To gather a sufficient number of tweets related to forced migration, the Twitter crawler uses Twitter's Search Tweets: Standard v1.1 (Access Date 1 June 2022). (https://developer.twitter.com/en/docs/twitter-api/v1/tweets/search/api-reference/get-search-tweets), which grants real-time access to public posts on the platform that include specific keywords or key phrases. The crawling iterations resulted in a vast collection of tweets, which were curated to focus mainly on refugee events, inclusion, and racism. The tool has been officially released and is available online inside the Memory Center Platform (MCP) (https://mcpwebstart.net/Home) to experiment with; a subscription to MCP is mandatory and only valid for cultural institutions.

The YouTube crawler, on the other hand, was developed to accumulate a sufficient number of video entities or metadata that are pertinent to the requirements on specific topics related to forced migration. This sub-component supports and provides two useful steps: it wraps around the YouTube Search: list v3 API (https://developers.google.com/youtube/v3/docs/search/list) to retrieve and store the responses of video metadata based on exact keyword search queries and the declared number of videos that the user wishes to inspect and, subsequently, allows the user to check the validity and relevance of the metadata information and retrieve and store the actual video footage in a remote repository. This framework has been designed from scratch to administer videos entitled with all sorts of Creative Commons video licenses officially disclosed on the platform. Those tools were tested across several collections of keywords, which were gathered by cultural institutions experts. The aim of the construction of such collections was to be able to retrieve the most-relevant content from the platforms. The quantitative results on the utilization of both tools are showcased in Table 1.

**Table 1.** Twitter/YouTube collection of keywords/key phrases details.

| Collections Title | Keywords/Key Phrases Included | Number of Tweets Retrieved | YouTube Metadata Entries Retrieved | Videos Downloaded |
|---|---|---|---|---|
| Definitions of actors in the process of social cohesion | refugee, immigrant, foreigners, asylum seekers, forced displacement | 116,174 | 30 | 8 |
| Needs and rights of refugees | refugee needs, refugee rights, right to citizenship, refugee documents, right to work, right to food, access to health, access to education, access to housing | 352,012 | 36 | 10 |
| Discrimination in host society | social stigma, social exclusion, racism, racial stereotypes, discrimination, steal jobs, blaming refugees | 15,282 | 47 | 10 |
| Gender | transphobia refugee, LGBTQIA+ refugee, refugee women rights, transgender refugee, homophobia refugee, gay refugee | 496 | 0 | 0 |
| Cultural heritage | cultural backgrounds, refugee shared heritage, refugee memorial heritage, workshop refugee, project refugee, refugee inclusion, refugee culture, refugee tradition, documentary refugee, poem refugee, refugee cinema, refugee storytelling, refugee recipes, refugee science, refugee language | 3233 | 83 | 15 |
| Emotions | empathy refugee, compassion refugee, anger refugee, refugee nostalgia, fear refugee, emotional heritage | 1170 | 47 | 6 |

**Table 1.** *Cont.*

| Collections Title | Keywords/Key Phrases Included | Number of Tweets Retrieved | YouTube Metadata Entries Retrieved | Videos Downloaded |
|---|---|---|---|---|
| Geography | refugee Poland, refugee Italy, refugee Greece, refugee Spain, refugee Catalunya, crossing borders | 5233 | 46 | 1 |
| Trauma | refugee support group, refugee violence, refugee trauma, vulnerable refugee | 1980 | 40 | 10 |
| Society | social integration, social inclusion, racist society, refugee integration, refugee community, foreign community, social cohesion, refugee acceptance, refugee empowerment, refugee working class, intercultural refugee, refugee oppression, refugee resistance | 28,143 | 95 | 20 |
| Religion | Muslim minority, Islam refugee, hijab refugee, refugee mosque, Ramadan refugee | 7156 | 48 | 3 |
| Tools | refugee phones, refugee camera, refugee virtual, refugee twitter, refugee Facebook, refugee Instagram, refuge social media | 59,101 | 50 | 10 |
| History and memory | European memory, memorial heritage, civil war, colonialism, decolonize, collective memory, dictatorship, exile | 565,385 | 50 | 7 |
| Approach in working with refugees | refugee victimhood, refugee assimilationism, ethnicization, folklorization, exoticism, Eurocentric | 5246 | 48 | 10 |
| Displacement | refugee homeland, refugee fatalism, refugee journey, country nostalgia | 2238 | 50 | 6 |

Lastly, the website-focused crawler was designed to detect specific parts of a web page by focusing on technical elements following the paradigm of well-structured websites. Experts formed a list of 55 potential websites, based on the plurality of the relevant thematic multimodal content they had, to serve as sources for crawling, but only three were chosen based on legal issues on crawling and content reusability, the abundance of multimedia content, well-structured templates, and software compatibility. All initial 55 websites, which were gathered by cultural institution experts, were deemed to be of equal importance and relevance to the cause of creating immersive digital experiences; thus, the final selection might be considered unbiased. The quantitative results of the final three website choices are presented in Table 2 along with the total crawling time in seconds per source and the total pages retrieved from each website. This approach of gathering raw data on forced migration and refugees is what enables cultural institutions to create immersive digital experiences that raise awareness and promote empathy towards these critical issues. The total size of the data retrieved from the websites was 2.0311 MB of raw targeted textual content. For Twitter, we estimated that the daily average volume in MBs was 643.92 MB. The total number of tweets retrieved was 1,162,849 tweets. We focused mainly on textual content from Twitter, where the average volume of a tweet is approximately 7.52 KBs, resulting in 13,580.29 MBs of textual data. For YouTube, we downloaded 670 metadata entities and 116 youtube videos. The total youtube videos volume was 1.7 GBs. The problem we needed to address was to be able to retrieve public content available on the web that was relevant to the theme of the refugee crisis in Europe. This content was to be reused in order to create some immersive experience in the Memory Center Platform.

**Table 2.** Websites Collection.

| Websites | Short Description | Focus | Total Crawl Time (sec) | Total Pages |
|---|---|---|---|---|
| amnesty.org/en/ | World's largest human rights movement | News reporting/policy | 4.89 | 1 |
| digitalmeetsculture.net | Portal for gathering information about world digital culture | News reporting on digital culture | 374.72 | 51 |
| cultural-opposition.eu | Project funded by EU | Cultural opposition/socialism in Eastern Europe | 183.29 | 113 |

*3.2. Semantic Framework*

In addition to reusing resources, it was deemed necessary to semantically annotate online items and harmonize them with other data in the system. Users can also manually insert multimedia items. The core ontology employed was the Dublin Core (DC) [34], which was extended to meet the requirements, such as to address custom relations among objects. The complete ontology (https://github.com/estathop/SO-CLOSE_ONTOLOGY) is depicted in Figure 1. The "dcterms:" prefix indicates established concepts from the DC that were reused, while the "soclose:" prefix represents custom concepts developed to address specific needs.

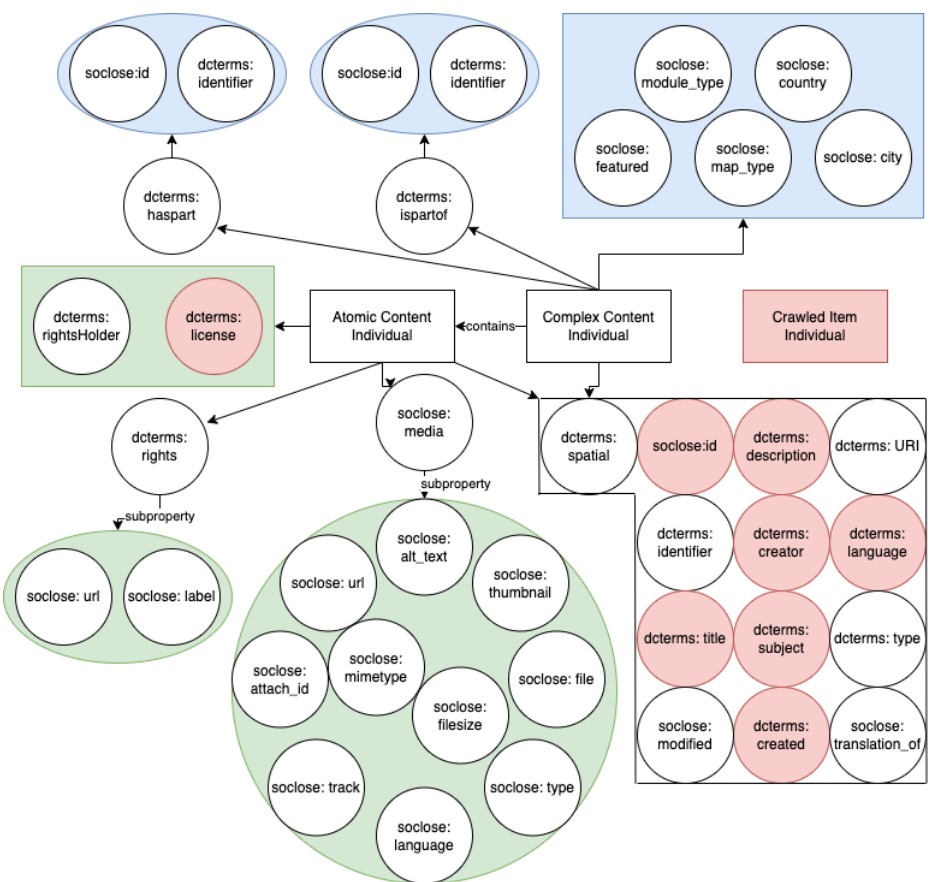

**Figure 1.** The SO-CLOSE ontology.

Aside from retrieved items, the system contains two other object types: atomic content and complex content. The atomic content item can be any form of multimedia (text, image, video, audio), while the complex content item necessarily consists of more than one atomic content item, forming story maps, virtual exhibitions, and web documentaries. Therefore,

both types share some common properties, which are displayed in the white-framed irregular hexagon in Figure 1. Each content type also has unique properties. The green shapes denote properties exclusive to atomic content instances, while the blue shapes are unique to complex content instances. For crawled item individuals (red rectangle) from websites and social media, only eight particular data type properties, depicted by the red color in Figure 1, were included to achieve semantic integration due to limitations in the provided metadata. Unnamed arrows indicate the ownership of the data type properties. Further information on the custom properties created can be found in Table 3, while information on the Dublin Core terms can be found in [34]. Automated semantic annotation of atomic and complex content items is performed as the framework extracts information from a platform where users input content manually. Crawled items are mapped on-demand from an internal MongoDB database, non-accessible for exterior use, where data are automatically stored from the web and social media.

**Table 3.** SO-CLOSE custom ontology properties.

| Name of Property | Short Description | Value Type |
|---|---|---|
| soclose:id | Internal database Identifier (ID). Only for internal use. | Non-negative integer |
| soclose:label | Label of the rights. | String |
| soclose:url | URL where the rights are described. | String (URL) |
| soclose:media | Information related to the media file. | Object |
| soclose:alt_text | Short phrase describing the image's purpose. | String |
| soclose:thumbnail | URL of a thumbnail image of the media file. | String (URL) |
| soclose:attach_id | Internal unique identifier of the media file. | Non-negative integer |
| soclose:mimetype | Mime type of the media file. | String |
| soclose:filesize | Media file size in number of bytes. | Non-negative integer |
| soclose:track | List of track of the media file (captions, descriptions, or transcriptions). | Array of objects |
| soclose:type | Identifies the type of track. | Caption, description, or transcription |
| soclose:language | Identifies the language of the track. | String (2-letter standard code) |
| soclose:file | The URL of the track's file. | String (URL) |
| soclose:translation_of | Internal ID of the original resource when the current resource is a translation. | Non-negative integer |
| soclose:modified | Last modification date of the resource. | Date in format YYYY-MM-DDThh:mm:ssTZD (ISO 8601) |
| soclose:featured | URL to the featured image for the exhibition. | String (URL) |
| soclose:map_type | Identifies the type of the map (only for stage and map). | String |
| soclose:country | Country (only for stage and map). | String |
| soclose:city | City (only for stage and map). | String |
| soclose:module_type | Type of module for stage, section, and panel. | textvideo, textimagebig, side2side, imagegallery, 3dgallery, v360, i360, vgallery, quote, map, juxtapose, textimage. |

Inference and Validation

Combining native OWL 2 Ruled-based Logic (RL) reasoning based on the OWL 2 RL profile semantics (OWL 2 RL/RDF rules [35]) resulted in additional logical assumptions. The semantic component enables the enrichment of relations among entities by using the CONSTRUCT graph pattern to support domain rules residing on top of the knowledge graph, thus identifying extra inferences. For example, when a complex content individual

pertains to a specific city and includes several atomic content individuals, the property "soclose:city" is attributed to all of them when the rule is triggered automatically, as demonstrated below:

```
CONSTRUCT {
 ?atomic_content soclose:city ?city
} WHERE {
 ?complex_content soclose:city ?city.
 ?atomic_content dcterms:haspart ?complex_content
}
```

To ensure quality in all aspects, both syntactical and morphological, a validation checking of the semantics was performed. This was achieved by using manually constructed SHACL validation rules (a language for validating RDF graphs against a set of conditions [36]) and native semantic consistency checking. Validation was achieved by considering the semantics at the terminological level, such as class disjointedness, while the first distinguishes constraint problems such as imperfect information or cardinality contradictions. For example, an SHACL shape guarantees that all complex content items have at least one atomic content as a part, as illustrated below:

```
soclose:ConShape
 a sh:NodeShape;
 sh:targetClass soclose:complex_content;
 sh:property [
  sh:path dcterms:haspart
  sh:minCount 1;
].
```

## 4. Methodology

In our research, as expounded in our previous work [37], we meticulously curated collections of keywords and key phrases with the help of cultural experts. These collections were centered on forced migration, human rights, and discrimination, aiming to retrieve relevant multimedia content.

Our primary objective in this paper was to assess various off-the-shelf query expansion techniques, measuring their effectiveness in retrieving pertinent multimedia from the domain of the refugee crisis. We believe the insights gleaned from this domain could also extend to other areas of interest, subject to further investigation. Furthermore, we examined the interoperability of these approaches with the semantic framework elucidated in Section 3.

We limited our focus to the most-prominent keyword from each of the 14 collections—the keyword responsible for retrieving the most-comprehensive online content, as documented in [37]. Our rationale here was to ensure ample content availability online for benchmarking query expansion, thus mitigating any potential bias from data scarcity.

The spaCy library's word embeddings (https://spacy.io/), in particular the "lg" model, were employed in conjunction with the cosine similarity to determine the distance between the initial phrase and the query expansion phrases across all techniques. We selected this model and library due to their high-quality pre-trained embeddings and seamless pipeline integration.

The evaluation metrics employed in this study included nDCG Equation (1) and MAP Equation (4). We first conducted a control execution without query expansion to calculate these metrics, subsequently repeating the process with query expansion terms. For the cosine similarity metric, a default threshold was adopted to classify an item (a video or a tweet post) as relevant or irrelevant to the initial key phrase based on its textual content.

This threshold, set at 0.5, is an empirically established value that has shown effectiveness in numerous applications [38].

$$DCG_p = \sum_{i=1}^{p} \frac{2^{rel_i} - 1}{\log_2{(i+1)}} \tag{1}$$

Given an estimated list of cosine similarity scores with which items are flagged in a binary approach as relevant or irrelevant, in Equation (1), $DCG_p$ represents the DCG at position $p$ and $rel_i$ denotes the relevance score of the item at position $i$. The equation calculates the sum of the discounted relevance scores for all items up to position $p$. Afterwards, the list of scores is sorted in descending order, and the IDCG Equation (2) is calculated with a similar mathematical formula, but this time applied to the sorted list.

$$IDCG_p = \sum_{i=1}^{p} \frac{2^{rel_i^*} - 1}{\log_2{(i+1)}} \tag{2}$$

In Equation (2), $IDCG_p$ represents the IDCG at position $p$, and $rel_i^*$ denotes the relevance score of the ideally sorted item at position $i$. The equation calculates the sum of the discounted relevance scores for all items up to position $p$ in the ideal arrangement. Notice that the formula looks similar to the DCG, but with the relevance scores sorted in descending order to represent the ideal ranking.

$$AP = \frac{1}{\text{number of relevant documents}} \sum_{k=1}^{n} (P(k) \times rel(k)) \tag{3}$$

In Equation (3), $AP$ represents the average precision, $P(k)$ is the precision at cut-off $k$, $rel(k)$ is an indicator function equaling 1 if the item at rank $k$ is relevant, and 0 otherwise, which in this case, the relevance is calculated through the threshold set from the cosine similarity scores. The sum is computed over all $n$ documents in the result set.

$$MAP = \frac{1}{|Q|} \sum_{q=1}^{|Q|} AP_q \tag{4}$$

In Equation (4), $MAP$ represents the Mean Average Precision, $AP_q$ is the average precision for query $q$, and $|Q|$ is the total number of queries. The MAP is the average of the AP values for all queries in the set. All the metrics above are commonly used for evaluating the performance of information retrieval systems, such as search engines.

Our methodology utilized the GPT-v3 main models, specifically the davinci-003 and curie-001 generative models, with set parameters aimed at retrieving the most-related terms for each key phrase. We also incorporated established knowledge graphs, such as DBpedia and WordNet corpus, to garner relevant phrases. For each keyword/key phrase, we limited consideration to the top 5 candidates from each query expansion attempt.

The evaluation process also included pre-processing steps aimed at refining the raw data and improving the cosine similarity scores' calculation. Procedures involved the removal of unnecessary characters or sub-strings, stop-word removal using the default spaCy dictionary, and case standardization. Additionally, language filters were applied to retrieve content solely in English, and DBpedia URIs were formatted to retrieve only individual names as potential expansion key phrases.

Throughout the methodology, we encountered several challenges that required addressal. For instance, the response variability from generative AI models, such as ChatGPT, required specific formatting of the query to ensure relevant results. Additionally, the occasional lack of related results from DBpedia often left the expansion candidate list vacant, thereby impacting the resulting statistics.

By outlining our methodology in this explicit manner in Figure 2, we hope to provide comprehensive insights into the theoretical foundations of our work, the motivations

guiding our choice of query expansion techniques, and the metrics employed to evaluate their effectiveness.

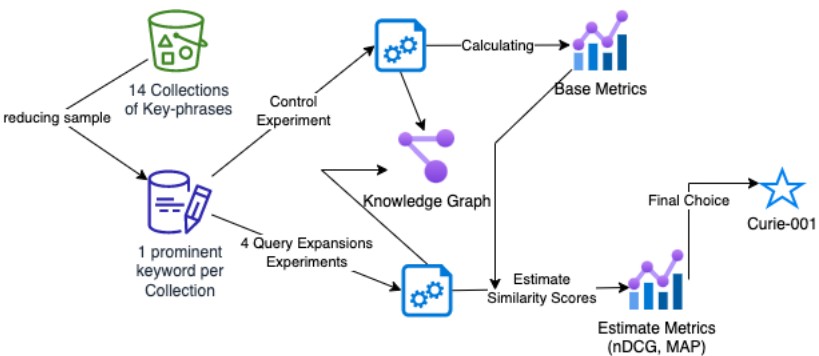

**Figure 2.** The SO-CLOSE experiment architecture.

## 5. Query Expansion Evaluation

Tables 4 and 5 contain the list of the initial key phrases along with some statistics. In these tables, the number of tweets/videos retrieved per each initial key phrase are shown with the maximum capacity set at 50 entities, along with how many tweets/videos passed the 0.5 cosine similarity threshold and were characterized as relevant. Additionally, the calculated values of the nDCG and MAP based on prior relevance characterization are included.

In Table 6, each key phrase, which was included in all evaluation trials, is demonstrated along with the calculated average cosine similarities in order to be able to compare each query expansion methodology. GPT-A refers to the davinci-003 generative model, and GPT-B refers to curie-001 generative model, whereas under DBpedia and WordNet, the equivalent average cosine similarities from the well-established knowledge graphs are showcased. It is evident that DBpedia under-performed at retrieving any content because, in eight cases, the knowledge graph when queried appropriately would not respond with any relevant query expansion material, thus scoring consecutive zeros on that metric. Among the other three techniques, the results of WordNet were substantially below those of the GPT models. The best two competitors were GPT-A and GPT-B; however, the average cosine similarity metrics of GPT-B outperformed in every key phrase when compared with any other method.

**Table 4.** Evaluation for key phrases related to refugees on Twitter before query expansion.

| Keyphrase | # of Tweets | # of Relevant Tweets | nDCG | MAP |
|---|---|---|---|---|
| Refugees crisis | 688 | 421 | 0.938 | 0.387 |
| Asylum seeker | 627 | 97 | 0.855 | 0.024 |
| Refugee needs | 274 | 213 | 0.937 | 0.581 |
| Social exclusion | 600 | 228 | 0.862 | 0.152 |
| Refugee rights | 561 | 387 | 0.942 | 0.431 |
| Refugee culture | 101 | 71 | 0.899 | 0.515 |
| Refugee storytelling | 6 | 3 | 0.928 | 0.317 |
| Empathy refugee | 28 | 13 | 0.907 | 0.19 |
| Crossing borders | 600 | 440 | 0.95 | 0.569 |
| Refugee violence | 224 | 185 | 0.933 | 0.686 |
| Social cohesion | 600 | 396 | 0.92 | 0.451 |
| Refugee acceptance | 14 | 10 | 0.895 | 0.55 |

**Table 5.** Evaluation for key phrases related to refugees on YouTube before query expansion.

| Keyphrase | # of Videos | # of Relevant Videos | nDCG | MAP |
|---|---|---|---|---|
| Refugees crisis | 50 | 39 | 0.965 | 0.647 |
| Asylum seeker | 50 | 35 | 0.915 | 0.507 |
| Refugee needs | 50 | 38 | 0.917 | 0.578 |
| Social exclusion | 50 | 42 | 0.875 | 0.732 |
| Refugee rights | 50 | 40 | 0.912 | 0.64 |
| Refugee culture | 50 | 38 | 0.892 | 0.533 |
| Refugee storytelling | 50 | 39 | 0.979 | 0.677 |
| Empathy refugee | 50 | 11 | 0.913 | 0.12 |
| Crossing borders | 50 | 42 | 0.887 | 0.646 |
| Refugee violence | 50 | 34 | 0.892 | 0.427 |
| Social cohesion | 50 | 46 | 0.978 | 0.903 |
| Refugee acceptance | 50 | 34 | 0.962 | 0.539 |

**Table 6.** Comparison of average similarity scores on different query expansion techniques.

| Keyphrase | GPT-A | GPT-B | DBpedia | WordNet |
|---|---|---|---|---|
| Refugees crisis | 0.516 | 0.858 | 0 | 0.434 |
| Asylum seeker | 0.345 | 0.572 | 0.285 | 0.132 |
| Refugee needs | 0.488 | 0.667 | 0 | 0.3 |
| Social exclusion | 0.685 | 0.876 | 0.401 | 0.388 |
| Refugee rights | 0.595 | 0.628 | 0.607 | 0.245 |
| Refugee culture | 0.706 | 0.725 | 0 | 0.413 |
| Refugee storytelling | 0.72 | 0.722 | 0 | 0.345 |
| Empathy refugee | 0.679 | 0.79 | 0 | 0.504 |
| Crossing borders | 0.464 | 0.656 | 0 | 0.287 |
| Refugee violence | 0.754 | 0.736 | 0 | 0.359 |
| Social cohesion | 0.716 | 0.734 | 0.574 | 0.352 |
| Refugee acceptance | 0.826 | 0.744 | 0 | 0.469 |

In Table 7, the final statistics for each query expansion method are illustrated. In particular, after the cosine similarity score was calculated per each distinct additional new key phrase, the average nDCG and MAP across all key phrases were calculated also per Twitter and YouTube. For Twitter, it is easily distinguishable that the GPT-A, GPT-B, and DBpedia expansion methods scored similarly high for the average nDCG for Twitter, but DBpedia outperformed all by far on the average MAP metric. Noticeably, WordNet was significantly worse on both the average nDCG and average MAP than the rest. For YouTube, the same differences as for Twitter applied, meaning that GPT-A, GPT-B, and DBpedia scored similarly well on the average nDCG, WordNet being the worst; however for YouTube, all four methods scored equally well on the average MAP also.

**Table 7.** Average nDCG and MAP metrics per expansion method for Twitter and YouTube.

| Expansion Method | Avg. nDCG (Twitter) [Std] | Avg. MAP (Twitter) [Std] | Avg. nDCG (YouTube) [Std] | Avg. MAP (YouTube) [Std] |
|---|---|---|---|---|
| GPT-A | 0.944 [0.008] | 0.297 [0.105] | 0.921 [0.017] | 0.459 [0.126] |
| GPT-B | 0.936 [0.009] | 0.223 [0.094] | 0.931 [0.020] | 0.497 [0.137] |
| DBpedia | 0.946 [0.007] | 0.471 [0.095] | 0.926 [0.019] | 0.472 [0.116] |
| WordNet | 0.885 [0.029] | 0.043 [0.144] | 0.845 [0.049] | 0.441 [0.147] |

In Table 4, which is for Twitter, the nDCG and MAP for "Social exclusion" are much lower than other key phrases such as "Refugee needs" and "Crossing borders". This indicates that the system is better at identifying relevant tweets for some key phrases than for others. Overall, the system achieved an average nDCG of 0.914 and an average MAP of 0.381 across all key phrases.

Looking at Table 5, it can be seen that the system performed relatively well on the key phrases, with an average nDCG of 0.922 and an average MAP of 0.583 across all key phrases. In terms of specific key phrases, the system achieved the highest nDCG and MAP for "Refugee storytelling", with scores of 0.979 and 0.677, respectively. On the other hand, the system performed the worst for "Social exclusion", with an nDCG of 0.875 and MAP of 0.732. Overall, the system achieved relatively high nDCG and MAP scores for the videos' data, indicating that it is effective at retrieving relevant videos for the given key phrases. However, it is worth noting that the performance varied significantly across key phrases, indicating that some key phrases are more challenging than others.

Finally, after calculating the metrics, it can bee declared that GPT-A, GPT-B, and DBpedia query expansion methods performed by three nDCG units better for Twitter and a little less better for YouTube. Regarding the average MAP, only DBpedia expansion worked better than the baseline for twitter, whereas all four methods scored lower for YouTube. As a result, it can be declared that, on DBpedia expansion, either highly related key phrases are retrieved or no key phrases at all. There is no supreme method over the rest, as it may depend on the key phrase, although GPT-B seemed to score a little better than the rest on most occasions.

## 6. Discussion

The practical repercussions of this study could fundamentally alter the landscape of data retrieval, analysis, and semantic integration, particularly in relation to socio-political issues such as forced migration and refugee crises. Our research advances the current state of the field by demonstrating the efficacy of our proposed methodology [39], opening up possibilities for a more refined and targeted approach to retrieving multimedia content across diverse online platforms. This innovation has a host of potential applications in both scholarly research and policy formulation settings.

Firstly, this research enriches existing work by equipping researchers, humanitarian organizations, and policymakers with a refined understanding of forced migration complexities and related concerns [40]. Leveraging state-of-the-art techniques for query expansion, keyword selection, and semantic integration [41], it becomes feasible to extract and analyze content that provides nuanced and precise insights. This advancement paves the way for more evidence-based decision-making, fostering the creation of more effective policies and intervention strategies.

Furthermore, our findings can inform the design and advancement of next-generation search engines and semantic repositories [42]. Through the integration of our query expansion techniques, evaluation metrics, and semantic framework, these platforms could provide end-users with more relevant results, thereby improving user satisfaction and engagement.

Moreover, organizations and companies working in social media analytics might find our proposed methodology intriguing. Applying our techniques [43] could allow these en-

tities to fine-tune their algorithms for more accurate sentiment and trend analysis related to various socio-political matters, which could guide their strategic decision-making processes.

A limitation observed in this study was the lack of automatic adaptability of the semantic framework, as the pipeline was tailored to cater to specific needs. For instance, if the Twitter API or YouTube Metadata API change, both the ontology and the entire framework would require modification, and sadly, these cannot be auto-generated or altered [44]. We minimized the bias during ontology engineering by consulting documented metadata from the APIs and using a widely-recognized generic ontology, such as the Dublin Core, extending it based on additional data in the system. Even with this strategy, it is important to note that ontology engineering does not lead to a single unique solution or outcome.

In conclusion, our research, while focused on the specific domain of refugee crises, has wider applicability and potential benefits. It can enhance the relevance and accuracy of data retrieval, improve the quality of information for decision-making, and contribute to the advancement of more sophisticated information retrieval systems and knowledge repositories. The real-world application opportunities of our findings are extensive, promising significant progress in the field of information retrieval, knowledge management, and data analysis.

## 7. Conclusions

In conclusion, this paper presented a query expansion benchmark on social media information retrieval, focusing on which methodology performs in alignment with semantic web technologies. Various query expansion techniques were examined, including query expansion with generative AI models such as GPT 3.0 and semantic matching algorithms that retrieve synonyms and similar phrases and keywords from well-established public knowledge graphs such as DBpedia and WordNet. The evaluation of these techniques included the use of cosine similarities to calculate the DCG value and the IDCG to finally calculate the nDCG, as well as the consideration of the mean average precision.

Furthermore, an analysis of the use of semantic web technologies as a component in a pipeline to receive and unify retrieved data from social media and build a thematic knowledge graph was presented and showed that those two different services can co-operate well as long as the same semantic layer is added on both. The raw material and metadata gathered can be used by cultural institutions to create immersive digital experiences about forced migration, as was explained thoroughly in the Introduction. Additionally, different frameworks were analyzed and presented in this paper, all of which function in harmony to retrieve key data with common semantic properties, along with unique information per source, to enhance and achieve ontological integration, which is feasible even with semi-structured data as long as pre-processing is present. The importance of semantically annotating online retrieved items and harmonizing them with other data inside the system using the Dublin Core (DC) ontology, which was extended to meet the requirements, was highlighted.

Moreover, automatic inference through CONSTRUCT graph patterns was presented along with validation checking of the semantics to ensure quality in all aspects, both syntactical and morphological, using manually constructed SHACL validation rules and native semantic consistency checking. That way, it was ensured that there were no logical or semantic violations within the populated knowledge graph and that knowledge expansion occurred with a correct approach.

Some practical contributions of this study are summarized on the following topics. The best-performing choice in query expansion techniques for Twitter and YouTube information retrieval among top generative AI models and well-known widely used knowledge graphs was identified through a standardized methodology presented in other works and which was based on statistical evaluation of well-established experiments. Additionally, a novel custom semantic framework including the expansion of a well-known ontology to

semantically annotate social media data and unify them automatically through a technical pipeline with other data present in the system was showcased.

Finally, the methodology that was followed to evaluate each query expansion technique was presented step by step by providing information about the mathematical metrics and the reasoning behind their choice and meaning. Established knowledge graphs, GPT models, and corpora for NLP were taken mainly into consideration to eliminate authors' bias and reassure integrity of the results. Necessary pre-processing steps, the choice of parameters, and challenges and difficulties were also presented. The proposed methodology, which relies on previous works, was evaluated, and the results were generated and illustrated in detail, as well as general assumptions.

Although the approach followed was methodic, some limitations appeared that might guide the research community for future research directions. There are numerous query expansions techniques as numerous generative AI models can perform this task. Moreover, only two widely used public knowledge graphs were utilized even though everyone could construct a knowledge graph and release it publicly. Therefore, initially, there was a matter of openness on linked data in knowledge graphs and the validity of the data themselves. Furthermore, ontologies and semantic annotation in general are not able to be generated or updated and exploited automatically if the data sources are changing either in structure or context, meaning that even a slight change in serving data through an API demands manual updates on the pipeline. Another limitation is that the similarity between expansion phrases and results was calculated automatically via an algorithm, whereas user-driven similarity might have yielded more intuitive results. Moreover, additional metrics usage and estimation might provide additional insights on such benchmarking procedures.

As future work, it is suggested to include more query expansion techniques, either more generative AI models or even conventional methods. Additional metrics estimated for IR could also provide further qualitative assessments when benchmarking different approaches. In terms of the semantic framework, the ontology can be expanded more in terms of concepts and properties so as to be able to be re-used in more generic use cases.

**Author Contributions:** The individual contributions of the authors are (according to CRediT taxonomy): conceptualization, methodology, and investigation, E.A.S. and A.I.K.; software, E.A.S., A.I.K., and A.K.; validation, E.A.S. and A.I.K.; formal analysis, E.A.S.; writing—original draft preparation, E.A.S.; writing—review and editing, E.A.S.; visualization, E.A.S. and A.I.K.; supervision, S.D., S.V., and I.K.; project administration, I.K. All authors have read and agreed to the published version of the manuscript.

**Funding:** This work was supported by the EC-funded project SO-CLOSE (H2020-870939).

**Institutional Review Board Statement:** Not applicable.

**Informed Consent Statement:** Not applicable.

**Data Availability Statement:** Due to the directive by the E.U. for anonymization of data and limitations of resources, data retrieved from social media data were deemed not to be available online. The custom ontology that was developed within the scope of this research was accessed at 9 June 2023 and can be found at: https://github.com/estathop/SO-CLOSE_ONTOLOGY.

**Conflicts of Interest:** The authors declare no conflict of interest. The funders had no role in the design of the study; in the collection, analyses, or interpretation of the data; in the writing of the manuscript; nor in the decision to publish the results.

## Abbreviations

The following abbreviations are used in this manuscript:

| | |
|---|---|
| DCG | Discounted Cumulative Gain |
| IDCG | Ideal Discounted Cumulative Gain |
| nDCG | normalized Discounted Cumulative Gain |
| AP | Average Precision |
| MAP | Mean Average Precision |

| IR | Information Retrieval |
|---|---|
| GPT | Generative Pre-trained Transformer |
| AI | Artificial Intelligence |
| RDF | Resource Description Framework |
| OWL | Web Ontology Language |
| SPARQL | SPARQL Protocol and RDF Query Language |
| NLP | Natural Language Processing |
| API | Application Programming Interface |
| DC | Dublin Core |
| RL | Rule-based Logic |
| SHACL | Shapes Constraint Language |

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
