# Peer review of "A Query Expansion Benchmark on Social Media Information Retrieval: Which Methodology Performs Best and Aligns with Semantics?"

_computers, doi:10.3390/computers12060119_

Round 1

Reviewer 1 Report

(1) The manuscript should provide more detailed descriptions of the datasets used in the experiments, including information on their size, diversity, and relevance to the research problem. This information is critical for assessing the generalizability of the results.

(2) The manuscript should also provide a more comprehensive discussion of the limitations and assumptions of the proposed methodology, especially in relation to the use of semantic web technologies. For instance, the manuscript could discuss the potential biases and inaccuracies that may arise from using existing ontologies or from the automatic generation of ontologies.

(3) The manuscript could benefit from a more detailed and explicit explanation of the theoretical underpinnings of the proposed methodology, including the assumptions and hypotheses that motivate the selection of the specific query expansion techniques and the evaluation metrics.

(4) The manuscript could further strengthen its contributions by discussing the practical implications of the findings, such as how the proposed methodology could be applied in real-world scenarios, and what benefits it could bring to end-users.

(5) Finally, the paper could benefit from a more systematic comparison with existing studies in the field, both in terms of methodology and results. This would help the authors to position their work in the broader context of social media information retrieval and query expansion research.

Overall, the language expression meets the quality requirements of the journal, but attention should be paid to the coherence of the logic before and after.

Author Response

Dear reviewer, see each number below according to each of your suggestions.

  1. Table 1 provides details on the Twitter and YouTube datasets regarding which key-phrases were selected by experts which fall under a certain thematic category of content that we wished to retrieve from the platforms. Additionally the site of the datasets is demonstrated in terms of numbers of items such as tweet post and youtube video entities on metadata and videos actually downloaded. On Table 2 the website sources are demonstrated regarding their URL, a short description representing their content and which is their main focus; we added total crawl times and total items crawled per source.  Lines 198 -207 added which elaborate more details about the datasets formed.
  2. The proposed methodology is based on previous published scientific work. For the part of the benchmark we followed the methodology and metrics in [3], [4], [5], [7], [8] and [9] and [23] which might not have been explicitly stated inside the chapter of methodology but it is assumpted from the related work we relied on. Lines 400-408 present limitations of the proposed framework.
  3. We restructed Chapter 5 entirely to make it more compact and concise. The revised passage provides a clearer explanation of the theoretical underpinnings, rationale for the choice of query expansion techniques, and an in-depth view of the evaluation metrics used in our study.
  4. Following this particular suggestion we added an additional chapter (“Discussions”) elaborating this comment.
  5. We have provided some minimally relevant references in the related work, due to the fact we were the first to try to benchmark these Generative AI models along with semantic algorithms on well-established knowledge graphs on socio-political crises. To our knowledge there is no such relevant bibliography trying to validate this kind of methodology in such as case, thus the lack of comparison. However we managed to add a few citations to justify motivation.

Reviewer 2 Report

This paper is recommended for publication in present form

Author Response

Thank you for your kind words

Reviewer 3 Report

This study has many scientific weaknesses in the literature review and the discussion of the results. Also, the organization of the manuscript is quite confusing.

Improvement suggestions:

1. The research gap is not addressed by the authors. The originally level of this study is not clear.

2. Introduction section needs to extended to justify the research gap.

3. Avoid colloquial tone like “In this Section, one can find…”

4. Motivation should be included in the Introduction section.

5. Motivation should be based in the literature and not in the personal views of the authors.

6. The collections title needs to be justified and properly framed in the literature.

7. Authors note “Experts formed a list of 55 potential websites to serve as sources for crawling, but only three were chosen based on legal issues on crawling and content reusability. Therefore, the risk of bias is very high.

8. Information provided in Table 2 is very scarce.

9. Authors mix the concept of methodology and presentation of results.

10. Statistical analysis provided in Table 7 is incomplete. Collect metrics regarding only average is not enough.

11. Authors don’t present a section to discuss the relevance of their findings.

12. Theoretical and practical contributions of this study are not address.

13. Limitations of their approach are not properly addressed.

14. The number of references is very scarce.

15. Some references present DOI, while others not. Provide DOI for all references.

Avoid the use of coloquial tone in some sentences. 

Author Response

  1. Reconstructed the methodology chapter entirely and added a discussion chapter in Section 7.
  2. Introduction was restructured entirely and refined.
  3. Rephrased colloquial tone where found such as in line 61 at the start of related work.
  4. As per this suggestion, motivation was fused with the introduction.
  5. Literature citations with DOIs (except for Tim-Berners Lee’s book) was added to the fused underlying motivation section inside the introduction .
  6. We added justification on the creation of those collections in lines 182-184 and 188-189
  7. We provide justification on why the the risk on bias is minimal on the selection of websites between lines 191-194.
  8. After this suggestion we added extra information on the total crawl time and the total number of web pages crawled to provide additional information on the website-focused crawler.
  9. In methodology chapter 5 we provided the basic notions, mathematics and theory in an abstract high level whereas while presenting the Query Expansion Evaluation Results we provided practical details on the materialization of the methodology illustrated in chapter 5. We believe that with this approach the paper appears more comprehensive to the reader and follows a more natural flow. However as performed in other chapters we tried to restructure the entire chapters.
  10. The aim of this paper was to provide a baseline initial benchmark based on averages (nDCG & MAP per Twitter & Youtube)  with multiple sources and tools being compared that was not done before in other related works. All relevant related work cited in this paper follow these particular metrics to acquire an intuition on the best performer, which is boardeline the curie-001 Generative AI model.  
  11. Following this suggestion we added a discussion section.
  12. Theoretical and practical contributions and future exploitation of the work performed in this paper is presented in Chapter 7 Discussion section.
  13. We added some limitations present to the current framework in Lines 400-408.
  14. We have provided additional relevant references in the introduction and related work to justify the purpose of this paper, the scope and the reseach gap it satisfies, however due to the fact we were the first to try to benchmark these Generative AI models along with semantic algorithms on well-established knowledge graphs and semantic framework on socio-political crises it was difficult to expand the literature. To our knowledge there is not much relevant bibliography trying to validate this kind of methodology in such as case.
  15. After your suggestion we managed to add lots of DOI numbers, however not all academic papers or resources may have DOI numbers, particularly older publications or those published in journals or proceedings that do not use DOI numbers. Also remember that some resources like arXiv preprints might not have DOI numbers, but instead have an arXiv identifier.

Round 2

Reviewer 1 Report

All the comments have been revised well.

Author Response

Sincerely thank you for your valuable feedback. 

Reviewer 3 Report

I acknowledge that the authors have made significant improvements in their work. However, several vulnerabilities remain that need to be better developed.

- The research gap needs to be better explored by the authors. It must be supported in the vision of up-to-date research studies in the theme.

- Literature review on semantic web technologies should be extended to provide a more detailed overview about the impact of these technologies in IR.

- It would be relevant to present a figure with the different phases of the methodological process.

- Table 6 should not be placed in the middle of a paragraph. Move it to its end.

- Statistical vision provided in Table 7 is very superficial. At least information regarding standard deviation should be provided.

- The discussion section is scientifically weak. References from previous studies should be used to demonstrate the advancements provided by this study.

- Conclusions section should also present the practical contributions provided by this study.

- Number of references is still low.

it can be improved.

Author Response

Initially, I would like to thank Reviewer 3 for his/her/them valuable feedback on suggesting alterations to upgrade the quality of the paper.

- The research gap needs to be better explored by the authors. It must be supported in the vision of up-to-date research studies in the theme.

We restructured and unified the chapter of related work by erasing the two subsections. We digged further into up-to-date research studies about information retrieval, query expansion and semantic knowledge graphs and managed to add 9 references with year of publication: 2014, 2 were published in 2018, 2019, 3 were published in 2020, 2021, 2023. Mainly lines 94-127. 8 out of 9 new entries had also a DOI to demonstrate. The research gap might also be better explained with additional 7 citations in the discussions sections where we analyzed how we built on top of existing works and solved problems that were present in the bibliography.

- Literature review on semantic web technologies should be extended to provide a more detailed overview about the impact of these technologies in IR.

As stated above, literature review was expanded with 9 extra up-to-date references, which followed information retrieval and semantic knowledge graphs. Mainly lines 94-127.

- It would be relevant to present a figure with the different phases of the methodological process.

After your suggestion we created and included a figure (figure 2) depicting the general experiment architectural flow.

- Table 6 should not be placed in the middle of a paragraph. Move it to its end.

The table was moved above to the previous page thus now it is not placed in the middle of the paragraph. It is accustomed first to present the object in a document then after provide its reference inside the text.

- Statistical vision provided in Table 7 is very superficial. At least information regarding standard deviation should be provided.

After this suggestion we also calculated and included the standard deviation per each expansion method and per each metric and source.

- The discussion section is scientifically weak. References from previous studies should be used to demonstrate the advancements provided by this study.

We added 7 references by previous studies in order to boost the discussion section scientifically and to showcase the advancements provided by this study in terms of trying to tackle existing problems in the literature.

- Conclusions section should also present the practical contributions provided by this study.

In Lines 474-481 we expressed the practical contributions of this study which can summarize to the best query expansion choice after testing and the semantic framework which unifies everything and forms a knowledge graph automatically.

- Number of references is still low.

We added 16 extra recent references and now the total citation number stands at 45, averaging 2,65 citations per page of the document.

Round 3

Reviewer 3 Report

The review carried out by the authors throughout this process was very positive. I just consider that the authors should address the limitations of their state in the Conclusions section. I would also recommend a better alignment of the limitations of the study with directions for future work. 

Quality of english is ok.

Author Response

"The review carried out by the authors throughout this process was very positive. I just consider that the authors should address the limitations of their state in the Conclusions section. I would also recommend a better alignment of the limitations of the study with directions for future work. "

We added lines 490-502 where current limitations of our work and research domain in general are presented. These limitations are in perfect alignment with the future work that is suggested in lines 503-507. Additionally we corrected some English errors to boost the quality of the language.